# CB1 Receptor Negative Allosteric Modulators as a Potential Tool to Reverse Cannabinoid Toxicity

**DOI:** 10.3390/molecules29081881

**Published:** 2024-04-20

**Authors:** Audrey Flavin, Paniz Azizi, Natalia Murataeva, Kyle Yust, Wenwen Du, Ruth Ross, Iain Greig, Thuy Nguyen, Yanan Zhang, Ken Mackie, Alex Straiker

**Affiliations:** 1Gill Center for Biomolecular Science, Program in Neuroscience, Department of Psychological and Brain Sciences Indiana University, Bloomington, IN 47405, USAnmuratae@iu.edu (N.M.); kyust@iu.edu (K.Y.); wd1@iu.edu (W.D.); kmackie@indiana.edu (K.M.); 2Department of Pharmacology and Toxicology, University of Toronto, Toronto, ON M5G 2C8, Canada; ruth.ross@utoronto.ca; 3Institute of Medical Sciences, University of Aberdeen, Aberdeen AB25 2ZD, UK; i.greig@abdn.ac.uk; 4Research Triangle Institute, Durham, NC 27709, USA; tnguyen@rti.org (T.N.); yzhang@rti.org (Y.Z.)

**Keywords:** cannabinoid toxicity, antidote, synthetic cannabinoid, overdose, JWH018

## Abstract

While the opioid crisis has justifiably occupied news headlines, emergency rooms are seeing many thousands of visits for another cause: cannabinoid toxicity. This is partly due to the spread of cheap and extremely potent synthetic cannabinoids that can cause serious neurological and cardiovascular complications—and deaths—every year. While an opioid overdose can be reversed by naloxone, there is no analogous treatment for cannabis toxicity. Without an antidote, doctors rely on sedatives, with their own risks, or ‘waiting it out’ to treat these patients. We have shown that the canonical synthetic ‘designer’ cannabinoids are highly potent CB1 receptor agonists and, as a result, competitive antagonists may struggle to rapidly reverse an overdose due to synthetic cannabinoids. Negative allosteric modulators (NAMs) have the potential to attenuate the effects of synthetic cannabinoids without having to directly compete for binding. We tested a group of CB1 NAMs for their ability to reverse the effects of the canonical synthetic designer cannabinoid JWH018 in vitro in a neuronal model of endogenous cannabinoid signaling and also in vivo. We tested ABD1085, RTICBM189, and PSNCBAM1 in autaptic hippocampal neurons that endogenously express a retrograde CB1-dependent circuit that inhibits neurotransmission. We found that all of these compounds blocked/reversed JWH018, though some proved more potent than others. We then tested whether these compounds could block the effects of JWH018 in vivo, using a test of nociception in mice. We found that only two of these compounds—RTICBM189 and PSNCBAM1—blocked JWH018 when applied in advance. The in vitro potency of a compound did not predict its in vivo potency. PSNCBAM1 proved to be the more potent of the compounds and also reversed the effects of JWH018 when applied afterward, a condition that more closely mimics an overdose situation. Lastly, we found that PSNCBAM1 did not elicit withdrawal after chronic JWH018 treatment. In summary, CB1 NAMs can, in principle, reverse the effects of the canonical synthetic designer cannabinoid JWH018 both in vitro and in vivo, without inducing withdrawal. These findings suggest a novel pharmacological approach to at last provide a tool to counter cannabinoid toxicity.

## 1. Introduction

While the opioid crisis has understandably occupied news headlines, emergency rooms have been seeing a ramp-up in visits due to another cause: cannabinoid overdose. This is partly due to the spread of cheap and extremely potent synthetic cannabinoids (SCs) that have resulted in many thousands of ER visits, serious neurological and cardiovascular complications, and roughly a dozen deaths per year [1]. For opioid overdose, there is naloxone, but no such tool is available for medical personnel confronting many thousands of cases of cannabinoid-related emergency department visits each year (reviewed in [2]).

Though not as widespread as cannabis, the more powerful designer SCs represent a more serious threat to individual health. Their availability and use have expanded rapidly, peaking in 2015, but they have persisted despite predictions that demand would wither with the legalization of cannabis. One attraction of these compounds is that they are likely to be missed by employer drug screens that may be enforced even in states that have legalized cannabis. For many years, SCs were synthesized based on the scaffold of JWH018, a potent analogue of THC, the primary psychoactive component of cannabis (reviewed in [3]). We characterized the neuropharmacology of JWH018 and several related compounds shortly after their appearance and showed that these compounds are higher-efficacy CB1 receptor agonists than THC [4,5]. Because SCs are more potent and efficacious than THC, their users often present to emergency rooms with tachycardia, vomiting, heart and kidney damage, seizures, and even deaths [1,6,7,8,9,10,11]. Synthetic cannabinoids initially benefitted from a legally ambiguous status; because they are chemically distinct from the psychoactive cannabinoid THC, they were not explicitly scheduled. For several years, these compounds were freely available, marketed as ‘legal weed’, and experienced an explosive growth in use. In response to the scheduling of first-generation SCs, drug developers rapidly developed third- and fourth-generation SCs that are structurally distinct and, thus, technically escape scheduling [3]. Though evidence suggests that these compounds have been selected for their activity at CB1 receptors, this may be changing [3].

Why is there no treatment for cannabis/cannabinoid overdose? There are no FDA-approved antagonists—no naloxone-equivalent—for CB1 receptors. One CB1 antagonist (rimonabant/Accomplia) was briefly approved in Europe for weight loss, but was withdrawn because long-term daily use was linked to dysphoric side effects (reviewed in [12]). Several pharmaceutical companies had active drug development programs with similar goals, but these were shut down when rimonabant was withdrawn. The drug industry has been reluctant to pursue further work with CB1 antagonists, even though a single acute treatment in an emergency setting represents a very different, and temporally constrained, therapeutic intervention. Clinical trials are under way for a signaling pathway-selective antagonist for cannabis use disorder [13], but this drug is intended for medium-term oral use rather than acute treatment in an emergency setting.

The promise of negative allosteric modulators: Because the newer synthetic cannabinoids are highly potent and efficacious agonists, there is a risk that a conventional competitive antagonist will be unable to rapidly reverse their effects due to their slow unbinding from CB1 receptors. This has proven to be an issue for the competitive mu opioid receptor antagonist naloxone in antagonizing synthetic opioids such as fentanyl and, especially, carfentanyl [14]. Competitive antagonists also bring the risk of inducing withdrawal, which may limit their acceptance. As depicted schematically in Figure 1, allosteric modulators act at a separate site and, thus, do not directly compete with a high-affinity orthosteric ligand. Allosteric modulators include several important classes of drugs, including benzodiazepines and barbiturates (at GABA_A_), and can act via several mechanisms, including changing the kinetics of orthosteric ligand binding [15]. Therefore, our hypothesis was that CB1 NAMs may be a preferred strategy to reverse cannabinoid toxicity, particularly in the face of an overdose, due to high-affinity synthetic cannabinoids. Here, we tested a group of promising CB1 NAMs both in vitro and in vivo for their ability to reverse the effects of JWH018.

We selected JWH-018 as the reference compound, since it was the original SC described in early preparations such as Spice and K2 and the structural well-spring for many subsequent SCs (Figure 2). The first research on SC ‘herbal blends’ in the club scene appeared in 2009 [16]. Producers of SCs busied themselves with selecting or designing other compounds, in large part in an effort to stay a step ahead of regulators and thereby continue the public sale of this class of compounds. However, there is a common structure to most of the synthetic cannabinoids that have been appropriated for SCs: a core heteroaromatic structure such as an indole ring (as in JWH018), a ‘head’ group consisting of a secondary structure (a naphthyl in JWH018) joining the core through a ‘bridge’ (carbonyl/methanone in JWH018), and a aliphatic/aromatic ‘tail’ at the N1 position of the core structure (a pentyl in JWH018).

## 2. Results

### 2.1. The Choice of CB1 NAMs and Synthetic Cannabinoid

As stated in the introduction, our objective was to test whether CB1 NAMs are able to reverse the effects of the potent high-affinity synthetic cannabinoid JWH018 in vitro or in vivo. To our knowledge, this has not previously been investigated. We have previously tested most of the first- and second-generation CB1 NAMs [17,18] and several CB1 antagonists in our neuronal model system and were, therefore, well-positioned to test these vs. synthetic agonists. In a previous test of a panel of CB1 NAMs [19], PSNCBAM1 was the clear standout and is included in this study. We also have evidence that the second-generation NAM ABD1085 ([20] an indole sulfonamide variant of ORG27569 with good oral potency) is effective in vivo [21] and included it for testing here. We additionally tested an arylurea-based NAM RTICBM189 [22]. We did not test pregnenolone or PEPCAN12 because we had found them to be either inactive or only slightly active in our model [17]. As noted in the introduction, these compounds were tested against the canonical synthetic designer cannabinoid JWH018.

### 2.2. CB1 Negative Allosteric Modulators Can Reverse JWH018 Effects on Cannabinoid Signaling in Autaptic Neurons

JWH018 decreases EPSC size in autaptic neurons. Antagonizing CB1 receptor activation would be expected to prevent this reduction in EPSC size and has been demonstrated in this model using negative allosteric modulators [17]. We treated neurons with JWH018 at 40 nM, a concentration that yielded a strong, but submaximal, inhibition of EPSCs in this model and measured the inhibition of EPSCs. We then added increasing concentrations of PSNCBAM1 to neurons treated with JWH018 to test whether the putative NAM could reverse the effects of JWH018. This is a requirement for a therapeutically useful compound in an emergency setting. We found that while PSNCBAM1 fully blocked DSE at 1 μM [19], it was unable to reverse the effects of JWH018 even at 2 μM, but was effective at 10 μM (Figure 3, relative EPSC charge after JWH018 ± SEM: 0.56 ± 0.07; after PSNCBAM1 (10 μM): 0.85 ± 0.05; *n* = 6; *, *p* < 0.05 one-way ANOVA with Dunnett’s posthoc test vs. JWH018). Thus, higher concentrations of PSNCBAM1 are needed to reverse the same extent of EPSC inhibition by JWH018, highlighting the possible probe dependence of PSNCBAM1, favoring the antagonism of 2-AG signaling.

We also tested ABD1085 [20], a candidate NAM that we have found effective in altering ocular pressure [21]. We had not previously tested this compound in the autaptic model and, thus, first tested the extent to which this compound might prevent DSE. We initially tested whether the compound had direct effects on neurotransmission by monitoring excitatory postsynaptic currents (EPSCs) before and after treatment with the compound. We found that a five-minute treatment with 1 μM ABD1085 did not alter EPSCs (Figure 4A; relative EPSC charge ± SEM: 0.99 ± 0.01, n = 5). We then elicited baseline DSE, followed by 5 min of treatment with ABD1085 followed by a second DSE in the presence of the drug, finding that 1 μM ABD1085 was able to diminish DSE in these neurons (Figure 4B,C). This concentration of ABD1085 was also able to reverse the effect of JWH018 (Figure 4D,E; relative EPSC charge after JWH018 ± SEM: 0.46 ± 0.02; after ABD1085 (1 μM): 0.97 ± 0.03; n = 5; *, *p* < 0.01 paired *t*-test vs. JWH018).

We also tested a 3-(4-chlorophenyl)-1-(phenethyl)urea analogue that has been proposed to act as a CB1 NAM. RTICBM189 has been shown to attenuate the reinstatement of cocaine-seeking behavior in a CB1-dependent manner [22]. We tested RTICBM189 for direct effects on neurotransmission and for its ability to reverse DSE in the autaptic model, finding that while RTICBM189 does not alter EPSCs on its own (Figure 5A, relative EPSC charge ± SEM: 1.05 ± 0.08, n = 5), it does reverse DSE at 1 μM (Figure 5B,C). RTICBM189 is also able to reverse the effect of JWH018 at this concentration (Figure 5D,E, relative EPSC charge after JWH018 (40 nM) ± SEM: 0.44 ± 0.05; after JWH018 + RTICBM189 (1 μM): 0.99 ± 0.06, n = 5, *p* = 0.006 by paired *t*-test).

From these experiments, we conclude that it is possible for a CB1 NAM to reverse the effects of JWH018 in a neuronal model of CB1 signaling.

### 2.3. CB1 NAMs Block Antinociception by JWH018 in a Mouse Model of Nociception

While it is encouraging that these compounds reverse JWH018 in vitro, ultimately, the key question is whether a CB1 NAM works in vivo. We tested whether the candidate NAMs reversed CB1-mediated JWH018 antinociception, since antinociception is among the best-studied cannabinoid effects. JWH018 antinociception can also be readily blocked or reversed, offering a real-time in vivo assay on the degree and time course of reversal of agonist effects. We used the tail flick assay, as it has previously been shown to be CB1-mediated [23,24]. We first obtained a dose–response curve for JWH018-induced antinociception by testing a range of concentrations (0.03–3 mg/kg). This allowed us to determine a dose (0.5 mg/kg) that results in a clear, submaximal effect (Figure 6A) that we then used in subsequent experiments. Candidate NAMs were first tested for their ability to block the effect of JWH018 when pre-injected (IP) 30 min before JWH018. Interestingly, while all compounds had proven effective in vitro, only two of the three blocked the effect of JWH018 in vivo. ABD1085 had been one of the more potent compounds tested in vitro, but did not affect antinociception produced by JWH018 (Figure 6B). RTICBM189 blocked the effect of JWH018 at 10 mg/kg (Figure 6C). PSNCBAM1 proved to be the most potent compound, reducing the effects of JWH018 even at 4 mg/kg, though not at 1 mg/kg (Figure 6D). Proceeding with PSNCBAM1, we tested whether the co-application of 10 mg/kg with JWH018 might also block the effects of the cannabinoid, finding that it did (Figure 6E). Lastly, we tested whether PSNCBAM1 could reverse established JWH018 antinociception, applying it 45 min after JWH018, finding that it rapidly reversed JWH018 antinociception, with clear effects as early as 15 min post-injection (Figure 6F).

### 2.4. PSNCBAM1 Does Not Induce Withdrawal

One limitation of conventional orthosteric antagonists is their precipitation of withdrawal in dependent individuals. This is particularly problematic in the case of opioids, where the sudden reversal of receptor activation induces powerfully aversive effects that can lead to aggressive, even violent, responses in victims who were at death’s door moments before. Negative allosteric modulators offer the promise of avoiding withdrawal because they modulate the activation of the receptor. We, therefore, tested whether PSNCBAM1 would induce withdrawal after 7 days of daily treatment with JWH018 (0.5 mg/kg, IP). PSNCBAM1 was administered (10 mg/kg, IP) 45 min after the final JWH018 treatment. Animals were monitored before and after PSNCBAM1 treatment for behaviors associated with cannabinoid withdrawal (e.g., head shakes and paw tremors). We found that PSNCBAM1 (10 mg/kg, IP) did not induce withdrawal, but the CB1 antagonist SR141716 (4 mg/kg, IP) applied afterward in a subset of animals did (Figure 7, withdrawal behaviors after treatment with JWH018 only (±SEM): 0.62 ± 0.26, n = 8; PSNCBAM1: 1.62 ± 0.86; n = 8; SR141716: 11.0 ± 3.6, n = 4; *p* < 0.005 for SR141716 vs. JWH018 via one-way ANOVA with Dunnett’s post hoc test).

## 3. Discussion

With tens of thousands of ER visits a year linked to cannabinoid overdose, there is a clear need for an antidote available to first responders [25]. An antidote against opioid overdose has served admirably in this role for more than 50 years, yet nothing is available for cannabinoid overdose. In principle, a competitive CB1 antagonist could serve this role; however, it may not be efficacious in an overdose of a high-affinity synthetic cannabinoid. An intriguing, novel approach to antagonizing CB1 signaling in an overdose setting would be to use a CB1 negative allosteric modulator. As a non-competitive agent, such a compound might prove to be versatile and effective against compounds regardless of their affinity at the orthosteric site. The current project was initiated to explore whether CB1 NAMs could effectively reverse the effects of an SC in vitro using a neuronal model of endogenous cannabinoid signaling and, if so, to learn whether such a compound might be effective in vivo. Our chief findings are that all CB1 NAMs tested were able to block and reverse the effects of JWH018 in autaptic neurons, though the potency varied considerably. Of the three compounds that were effective in vitro, only two were able to prevent the effect of JWH018 when given in advance. Only one of these—PSNCBAM1—was effective at less than 10 mg/kg. Importantly, PSNCBAM1 was also able to reverse the effects of JWH018 when given after JWH018, mimicking an emergency situation. Importantly, PSNCBAM1 did not induce withdrawal in mice chronically treated with JWH018. Taken together, our findings suggest that a CB1 NAM can, in principle, serve as an antidote to cannabinoid overdose/toxicity in an emergency setting.

The successful use of an antidote in an emergency setting comes with some specific requirements. First, the compound must be effective even against a high-affinity compound. Our tests against the canonical SC JWH018 suggest that a CB1 NAM can work against a high-affinity agonist in vivo. Second, the antidote must be fast-acting, preferably in minutes. Here, we saw evidence for an effect within 15 min after IP injection, and it is possible that the compound acts more rapidly. Most CB1 NAMs described so far have lipophilic structures and, thus, are expected to readily penetrate the blood–brain barrier. However, this may be affected by the route of administration, and this is the third major requirement: the method of administration must be suited to an emergency setting. SCs have been associated with psychotic episodes, meaning that treatment may not be voluntary. In emergency situations outside of a hospital, intranasal and intramuscular (IM) modes of treatment are the most likely methods of administration, but for an uncooperative patient, IM injection may be the only suitable approach. Lipophilic compounds applied intramuscularly may have longer loitering times, since they readily absorb into local membranes. The presence of adipose tissue may also serve as a local sink for injected compounds. The question of how rapidly a compound such as PSNCBAM1 works will require pharmacokinetic studies in human subjects.

One interesting aspect of our findings was that the potency of drugs in the autaptic model was not a predictor of the effectiveness of a compound in vivo. Indeed, the least potent compound in terms of reversing the effects of JWH018 in autaptic neurons proved most potent in vivo. It is possible that CB1 antinociceptive effects are less dependent on the signaling pathways that underlie the inhibition of calcium channels in neurons that form the basis of DSE [26]; however, the difference may have resulted from the pharmacokinetic properties of these different CB1 NAMs, such as their ability to pass across the blood–brain barrier. While ABD1085, an indole derivative containing a sulfonamide group, was shown to be brain-penetrant in in vitro permeability and in vivo pharmacokinetic studies, it has a T_max_ of 4 h when orally dosed, i.e., it does not reach maximum concentration until 4 h after administration [20]. In contrast, RTICBM189, an arylurea-based CB1 NAM, has a T_max_ of 0.4 h when IP administered [22]. This may explain why RTICBM189 was active against JWH018 with 30 min pre-administration, while ABD1085 had no effect. To the best of our knowledge, PSNCBAM1 has not been tested for its in vivo pharmacokinetic properties. The fact the PSNCBAM1 was active against JWH018 during 30 min pre-treatment, co-administration, and 45 min after JWH018 injection in the tail flick assay suggests that it may have more rapid brain penetration, although this will need to be experimentally confirmed.

This also raises another important consideration: what is a CB1 antidote reversing? For an opioid antagonist, the picture is clear, since deaths associated with opioid overdose follow severe respiratory depression. A successful opioid antidote should, therefore, at a minimum, reverse respiratory depression. But in the case of CB1 overdose, the situation is less clear. First, the great majority of hospital visits are due to cannabis toxicity rather than overdose from synthetic cannabinoids. The symptom profiles for these are different, though it is likely that both are largely due to CB1 activation. According to a recent study of children admitted to emergency rooms, the most common symptoms were slowed breathing and reduced heart rates [25].

Individuals experiencing cannabinoid overdose frequently experience agitation, anxiety, and vertigo, but may report a wide range of symptoms. The effects may also vary depending on the route of entry (i.e., inhaled vs. ingested), adulterants that may be present, and also may depend on individual sensitivity. There is evidence that populations with cardiovascular conditions may be vulnerable to the effects of cannabinoids (e.g., [27]). The cardiovascular effects likely tie into some of the symptoms being seen with synthetic cannabinoid overdoses, such as tachycardia. While we have convincing animal data demonstrating the in vivo effect of PSNCBAM1, the current study is limited to only its anti-nociceptive effect, and it is possible that the same compound is less effective for other physiological consequences of cannabinoid toxicity. Thus, future studies should determine the efficacy of PSNCBAM1 to reverse other preclinical signs of JWH018-mediated CB1 activation such as hypothermia, catatonia, etc.

A second challenge in developing an antidote to synthetic cannabinoids has to do with the nature of these synthetic cannabinoids. For cannabis toxicity, the chief concern is THC, mainly acting at CB1 receptors in the CNS. Over the last dozen years, it was safe to assume that CB1 was the chief target of synthetic cannabinoids, but there are signs that this may be changing. Because government regulations tended to be reactive, targeting specific compounds once they appeared on the drug scene, producers focused on ‘tweaking’ the original compound to develop new drugs that would evade restrictions while preserving the desired CB1 agonist profile. More than 200 synthetic cannabinoids have appeared over the last 15 years. However, the Chinese government—and there is evidence that the bulk of these compounds are produced in China—has enacted legislation targeting the scaffold itself, with the consequence of a shift to different scaffolds (reviewed in [3]), a strategy known as scaffold-hopping. There is a risk that these will have a much different pharmacological profile (i.e., they may engage other CNS receptors in addition to CB1), though it seems likely that the compounds will still have strong CB1 agonist properties. While previous compounds reliably served as strong, relatively selective CB1 agonists and, thus, motivated the development of a CB1 antidote, the current study only examined JWH018, and this may not hold for future classes of SCs.

Our finding that, in contrast to the competitive CB1 antagonist SR141716A, PSNCBAM1 did not induce withdrawal may have broad implications for the therapeutic application of CB1 NAMs. Given the sheer numbers of opioid-related overdoses and fifty years of experience with naloxone, there is considerable experience with opioid withdrawal in an emergency setting. Opioid withdrawal is deeply aversive and there are countless reports of aggression and even violence against first responders during precipitated opioid withdrawal. For cannabinoid toxicity, to our knowledge, this has not been investigated in humans, since no CB1 antagonist has been available for this purpose. There is evidence for a spontaneous cannabis withdrawal syndrome with symptoms such as anxiety and irritability [28]. However, precipitated withdrawal in preclinical models of cannabinoid dependence is accompanied by strong signs, reminiscent of opioid withdrawal in similar models. Thus, if a compound serves as an antidote to cannabinoid toxicity without inducing withdrawal symptoms, this would presumably be advantageous over one that does. On a related point, it is possible that a CB1 NAM would not cause the dysphoric effects that led to the withdrawal of rimonabant (SR141716A) (reviewed in [12]). Those effects were presumably due to the action of rimonabant as an inverse agonist at CB1 and, thus, are unlikely to be mimicked by a CB1 NAM. In any event, a single use in an emergency setting may allow for some more flexibility in terms of the side effect profile.

With hundreds of thousands of emergency room visits related to cannabinoid toxicity each year, the need for an antidote is compelling, but the rimonabant experience has long chastened pharmaceutical companies from working in this space. To fill this void, we have tested a panel of CB1 negative allosteric modulators for their ability to reverse the effects of a canonical designer synthetic cannabinoid, JWH018. We found that all CB1 NAMs were able to reverse the effects in vitro, and that two of the compounds acted similarly in vivo. One of these, PSNCBAM1, was additionally tested and found to reverse JWH018 effects and was found to act without inducing withdrawal in a mouse model. Our findings serve as proof-of-concept that a CB1 NAM could serve as an antidote for cannabinoid toxicity, though additional studies will be required to further assess the suitability of this strategy.

## 4. Materials and Methods

### 4.1. Animals

Adult male wild-type mice with a C57BL/6J background were purchased from The Jackson Laboratory (Bar Harbor, ME, USA) and used in behavioral studies. Females were not included for this testing because it represents the first step and a proof-of-concept study of whether CB1 NAMs can, in principle, impact JWH018 responses in vitro and in vivo. Future studies to develop compounds as antidotes for cannabinoid toxicity will test both males and females. For neuronal culture, mouse pups (age postnatal day 0–2, indeterminate sex) were used. Seventy-two mice were used for this study. All mice were ~12–20 weeks old when used in this study. All mice were maintained on a 12 h reverse light/dark cycle (lights off from 8 a.m. to 8 p.m.) in a temperature- and humidity-controlled facility and allowed ad libitum access to food and water throughout the experimental period. All experiments were approved by the Indiana University Bloomington Animal Care and Use Committee.

### 4.2. Hippocampal Culture Preparation

Mouse hippocampal neurons isolated from the CA1–CA3 region were cultured on microislands as described previously [29,30]. Neurons were obtained from animals (age postnatal day 0–2) and plated onto a feeder layer of hippocampal astrocytes that had been laid down previously [31]. Cultures were grown in high-glucose (20 mM) DMEM containing 10% horse serum, without mitotic inhibitors, and used for recordings after 8 days in culture and for no more than 3 h after removal from the culture medium.

### 4.3. Electrophysiology

When a single neuron is grown on a small island of permissive substrate, it forms synapses or ‘autapses’ onto itself. All experiments were performed on isolated autaptic neurons. Whole-cell voltage-clamp recordings from autaptic neurons were carried out at room temperature using an Axopatch 200 A amplifier (Molecular Devices, Sunnyvale, CA, USA). The extracellular solution contained (in mM) 119 NaCl, 5 KCl, 2.5 CaCl_2_, 1.5 MgCl_2_, 30 glucose, and 20 HEPES. The continuous flow of solution through the bath chamber (~2 mL/min) ensured rapid drug application and clearance. The drugs were typically prepared as stocks, and then diluted into extracellular solution at their final concentration and used on the same day.

Recording pipettes of 1.8–3 MΩ were filled with (in mM) 121.5 KGluconate, 17.5 KCl, 9 NaCl, 1 MgCl_2_, 10 HEPES, 0.2 EGTA, 2 MgATP, and 0.5 LiGTP. Access resistance and holding current were monitored and only cells with both stable access resistance and holding current were included for data analysis. The conventional stimulus protocol was as follows: the membrane potential was held at −70 mV and excitatory postsynaptic currents (EPSCs) were evoked every 20 s by triggering an unclamped action current with a 1.0 ms depolarizing step. The resultant evoked waveform consisted of a brief stimulus artifact and a large downward spike representing inward sodium currents, followed by the slower EPSC. The size of the recorded EPSCs was calculated by integrating the evoked current to yield a charge value (in pC). Calculating the charge value in this manner yields an indirect measure of the amount of neurotransmitter released while minimizing the effects of cable distortion on currents generated far from the site of the recording electrode (the soma). Data were acquired at a sampling rate of 5 kHz.

DSE stimuli: After establishing a 10–20 s 0.5 Hz baseline, DSE was evoked by depolarizing to 0 mV for 3 or 10 s, followed by the resumption of a 0.5 Hz stimulus protocol for 20–80 s, allowing EPSCs to recover to baseline values. To allow comparison, baseline values (prior to the DSE stimulus) were normalized to one. DSE inhibition values are presented as fractions of 1, i.e., a 50% inhibition from the baseline response is 0.50 ± standard error of the mean.

### 4.4. Tail Flick Assay

To assess the latency of an evasive response to a nocifensive stimulus that is sensitive to cannabinoid CB1 receptor activation [23], we used the tail flick assay. Mice were gently handled by the experimenter several days before the testing. Prior to the experiment, mice were introduced to the environment for 30 min for adaptation. At the outset of the experiment, three separate baseline values were recorded (with a 10 min interval between each). A cut-off of 15 s was applied to avoid tissue damage. If a mouse did not exhibit a nocifensive response by the cut-off time, the test was terminated, a latency of 15 s was recorded, and the mouse was defined as analgesic.

To determine a dose response for JWH018 in this model, naïve mice were exposed to an acute dose of JWH018 (0.03–3 mg/kg; i.p.) and subsequently tested for latencies every 15 min over 75 min post-injection. Based on the dose response, we chose 0.5 mg/kg JWH018 for subsequent experiments.

Candidate CB1 NAMs were tested for their ability to mitigate JWH018-induced antinociception. Compounds were initially tested at 10 mg/kg (i.p.) administered 30 min before JWH018. Following this preliminary assessment, PSNCBAM1 was selected for further testing. This involved the co-application of PSNCBAM1 and JWH018 and the monitoring of nociceptive responses as above. Additionally, in a separate experiment, PSNCBAM1 was administered 45 min after JWH018 to determine whether it would antagonize antinociception with JWH018.

### 4.5. Withdrawal Behavior

Mice were injected daily with JWH018 (0.5 mg/kg, IP) for one week. Forty-five minutes after the seventh injection, mice were treated with PSCNBAM1 (10 mg/kg, IP). Mice were monitored for 15 min after the last JWH018 treatment starting 15 min after the injection and again 15 min after PSNCBAM1 injections. Head shakes and paw tremors were counted. A subset of mice additionally received an injection with SR141716 (4 mg/kg, IP) 45 min after their PSNCBAM1 injections and were monitored as above for evidence of withdrawal behaviors.

### 4.6. Drugs

JWH018 was purchased from Sigma-Aldrich (St. Louis, MO, USA). ABD1085 was provided by Dr. Iain Greig. RTICBM189 was provided by Dr. Yanan Zhang. PSNCBAM1 was purchased from Cayman Chemical (Ann Arbor, MI, USA). For electrophysiology experiments, drugs were prepared as a stock at a high concentration (typically 10 mM) in DMSO or ethanol, then diluted shortly before a given experiment. Solvent concentrations did not exceed 0.1%. For in vivo experiments, drugs were prepared for intraperitoneal injection in an ethanol/kolliphor/saline mixture (1:1:18). In the case of JWH018, which was received from the manufacturer in methanol, the methanol was evaporated away shortly before the experiment and resuspended in ethanol, then added to the kolliphor/saline mixture as above.

## Figures and Tables

**Figure 1 molecules-29-01881-f001:**
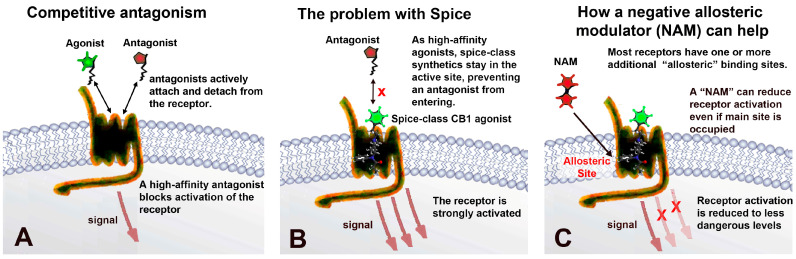
Negative allosteric modulators of CB1 receptors and their potential to reverse acute synthetic designer cannabinoid-mediated toxicity. (**A**) Competitive antagonists compete directly at the orthosteric site. (**B**) Powerful synthetic “spice-class” cannabinoids may bind tightly to the orthosteric site and strongly activate the receptor. This tight binding may prevent a competitive antagonist from reversing the effect (indicated by red cross). (**C**) A negative allosteric modulator targets a secondary ‘allosteric’ site, reducing the activity of the receptor (red crosses) without directly competing at the orthosteric site.

**Figure 2 molecules-29-01881-f002:**
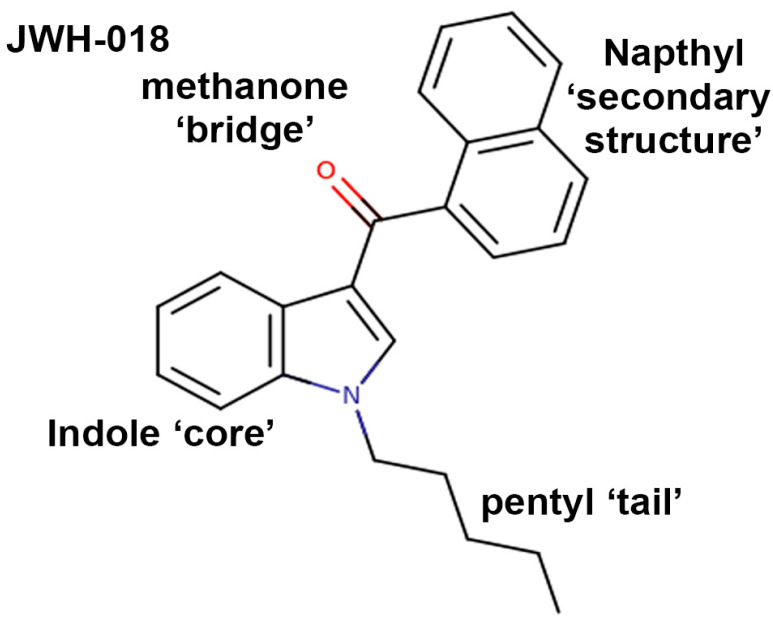
Structure of JWH-018 progenitor synthetic designer cannabinoid.

**Figure 3 molecules-29-01881-f003:**
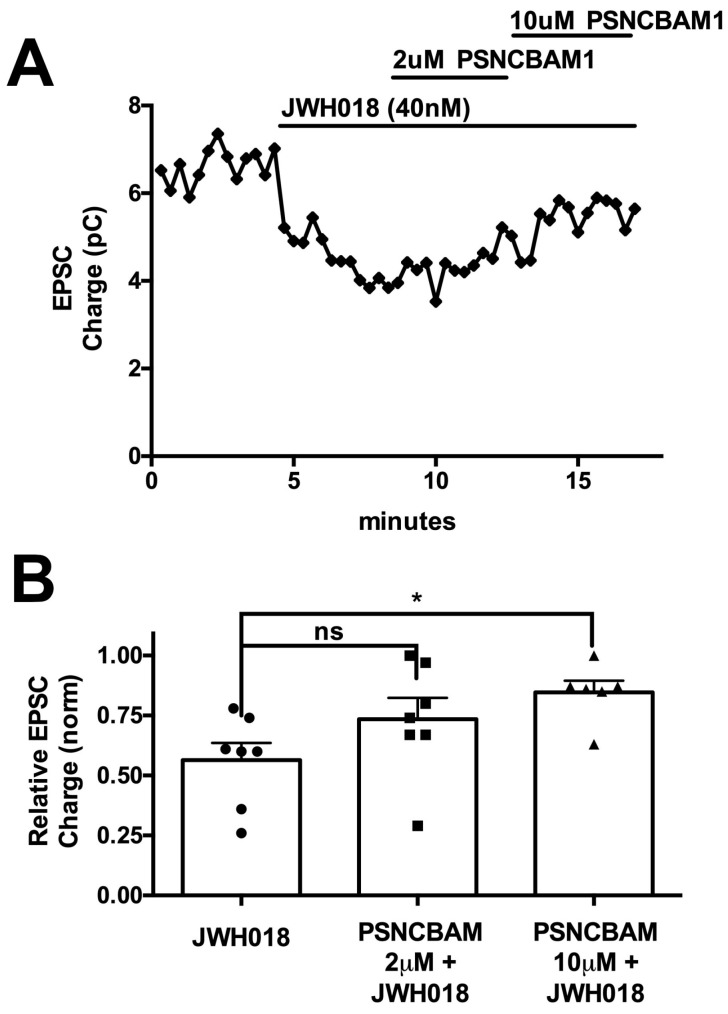
A high concentration of PSNCBAM1 reverses JWH018 inhibition of neurotransmission. (**A**) Sample time showing inhibitory effect of JWH018 (40 nM) and partial reversal by PSNCBAM1. (**B**) Summary bar graph shows a partial reversal of JWH018 (40 nM) effect on EPSCs (circles) by PSNCBAM1 at 10 μM (triangles), but not 2 μM (squares). ns, not significant; *, *p* < 0.05, by one-way ANOVA with Dunnett’s post hoc vs. JWH018.

**Figure 4 molecules-29-01881-f004:**
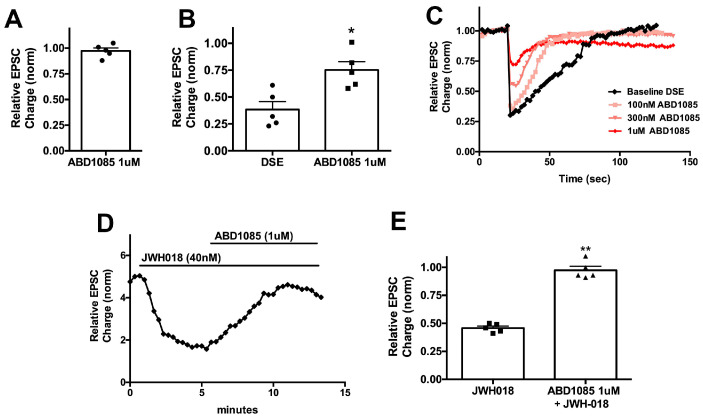
ABD1085 inhibits DSE and reverses JWH018 effects on neurotransmission in autaptic neurons. (**A**) Negative allosteric modulator ABD1085 has no effect on EPSCs at 1 μM. (**B**) ABD1085 (squares) partly blocks DSE (circles) at 1 μM. (**C**) Sample DSE time courses after treatment with increasing concentrations of ABD1085. (**D**) Sample time course shows a full reversal of JWH018 (40 nM) effect on EPSCs by ABD1085 (1 μM). (**E**) Summary graph shows EPSC inhibition by JWH018 (squares) and reversal by ABD1085 (1 μM, triangles). *, *p* < 0.05 by paired *t*-test. **, *p* < 0.01 by paired *t*-test.

**Figure 5 molecules-29-01881-f005:**
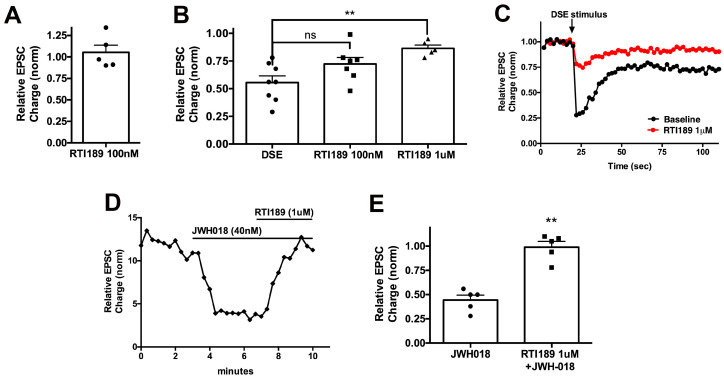
RTICBM189 inhibits DSE and reverses JWH018 effects on neurotransmission in autaptic neurons. (**A**) Negative allosteric modulator RTICBM189 has no effect on EPSCs at 1 μM. (**B**) RTICBM189 does not block DSE (circles) at 100 nM (squares), but does at 1 μM (triangles). (**C**) Sample DSE time courses before and after treatment with RTICBM189. (**D**) Sample time course shows a full reversal of JWH018 (40 nM) effect on EPSCs by RTICBM189 (1 μM). (**E**) Summary graph shows EPSC inhibition by JWH018 (circles) and reversal by RTICBM189 (1 μM, squares). ns, not significant; **, 0 < 0.01 one-way ANOVA with Dunnett’s post hoc test (**B**), by paired *t*-test (**D**).

**Figure 6 molecules-29-01881-f006:**
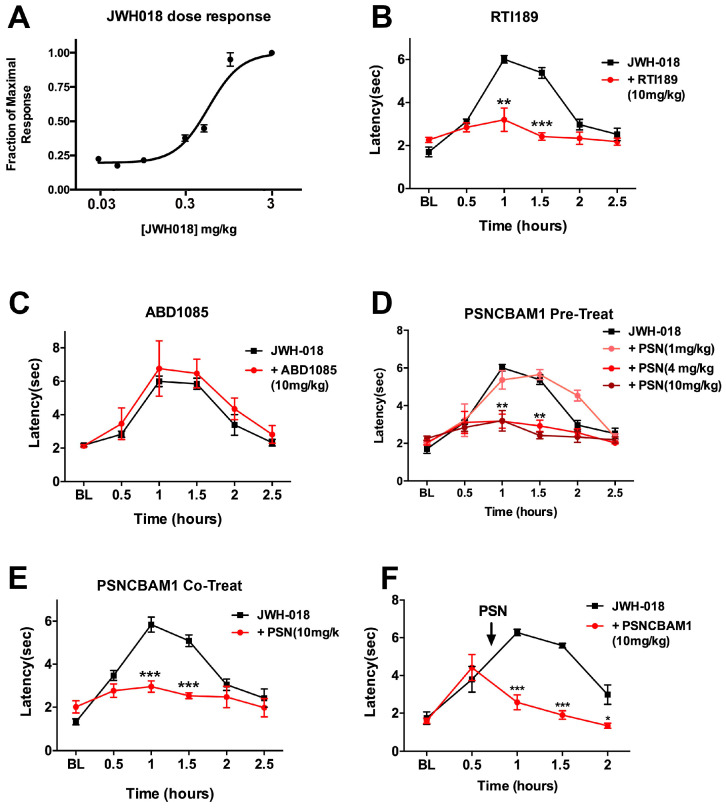
CB1 NAMs vs. JWH018 effects on nociception. (**A**) Dose response for JWH018 in tail flick assay. (**B**) Time course shows that pretreatment with RTICBM189 at 10 mg/kg blocks the effect of JWH018 (0.5 mg/kg). (**C**) ABD1085 is ineffective at 10 mg/kg. (**D**) Pretreatment with PSNCBAM1 blocks effect of JWH018 at 4 and 10 mg/kg. (**E**) PSNCBAM1 (10 mg/kg) is still effective when co-treated with JWH018 (0.5 mg/kg). (**F**) PSNCBAM1 (10 mg/kg) rapidly reverses the effect of JWH018 when applied 45 min after JWH018. *, *p* < 0.05, **, *p* < 0.01, ***, *p* < 0.005, two-way ANOVA with Bonferroni post hoc test.

**Figure 7 molecules-29-01881-f007:**
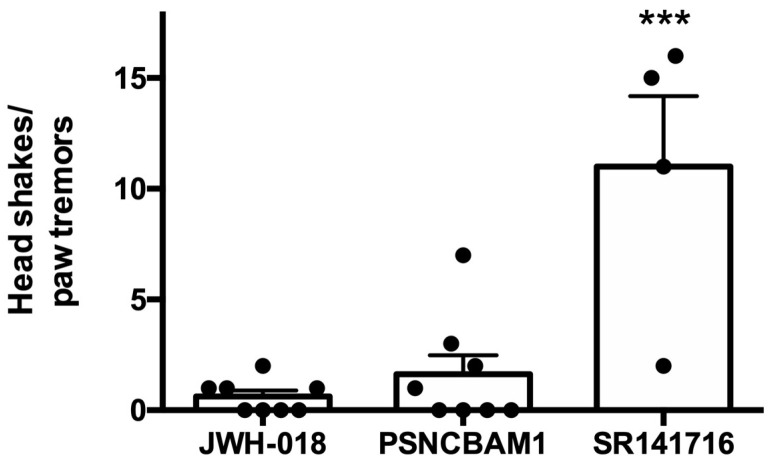
PSNCBAM1 does not induce withdrawal in JWH018-treated mice. Mice treated for 7 days with daily JWH018 injections (0.5 mg/kg) did not see withdrawal symptoms 45 min after treatment with PSNCBAM1 (10 mg/kg). A subset of mice subsequently treated with the competitive antagonist SR141716 showed an increase in withdrawal symptoms. ***, *p* < 0.005, one-way ANOVA with Dunnett’s post hoc test vs. JWH018.

## Data Availability

The data underlying this article will be shared on request to the corresponding author.

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
