# Peer review of "CB1 Receptor Negative Allosteric Modulators as a Potential Tool to Reverse Cannabinoid Toxicity"

_molecules, 2024, doi:10.3390/molecules29081881_

Round 1

Reviewer 1 Report

Comments and Suggestions for Authors

This manuscript describes the effects of several CB1R NAMs on cannabinoid JWH108-induced electrophysiological and behavioral effects, both in vitro neuronal cultures and in vivo. Overall, it is an interesting study with some novel findings. The manuscript is well-written. I have only a few comments to further improve this manuscript:

The title highlights the effects of those NAMs on cannabinoid toxicity, while the study focuses on their effects on neurotransmission and pharmacological analgesia. The authors did not provide evidence indicating that repeated JWH018 treatment alone or the NAM (PSNCBAM1) injection produced any neurotoxic behaviors such as withdrawal symptoms.

To address this question (which is very interesting), the authors should consider a series of new experiments to systematically evaluate whether such NAMs can reverse repeated THC- or JWH018-induced toxicity (if any) or rimonabant-precipitated withdrawal signs in THC- or JWH018-treated mice, such as CPA, jumping, diarrhea, head and body shakes, paw tremor, etc. Otherwise, the authors should revise the title and related statements in this manuscript.

JWH018 is not a selective CB1R agonist. Actually, it has high binding affinity to both CB1 and CB2 receptors with Ki values of 9 nM and 2.9 nM, respectively. As evidence has shown that hippocampal neurons also express functional CB2 receptors (referencing Stempel AV et al., 2016), it is necessary to evaluate whether pretreatment with CB1 or CB2 receptor antagonists will block the NAM action on JWH018-induced reduction in EPSCs.

Author Response

Reviewer #1

This manuscript describes the effects of several CB1R NAMs on cannabinoid JWH108-induced electrophysiological and behavioral effects, both in vitro neuronal cultures and in vivo. Overall, it is an interesting study with some novel findings. The manuscript is well-written. I have only a few comments to further improve this manuscript:

The title highlights the effects of those NAMs on cannabinoid toxicity, while the study focuses on their effects on neurotransmission and pharmacological analgesia. The authors did not provide evidence indicating that repeated JWH018 treatment alone or the NAM (PSNCBAM1) injection produced any neurotoxic behaviors such as withdrawal symptoms.

To address this question (which is very interesting), the authors should consider a series of new experiments to systematically evaluate whether such NAMs can reverse repeated THC- or JWH018-induced toxicity (if any) or rimonabant-precipitated withdrawal signs in THC- or JWH018-treated mice, such as CPA, jumping, diarrhea, head and body shakes, paw tremor, etc. Otherwise, the authors should revise the title and related statements in this manuscript.

A: The goal of the study was to test whether CB1 NAMs might serve as a tool for doctors to reverse the effects of cannabinoids.   And we should add that we did in fact test for withdrawal symptoms, showing that repeated treatments with JWH018 followed by rimonabant causes withdrawal symptoms (Fig 7) and that PSNCBAM1 does not.  However we have altered the title.  

JWH018 is not a selective CB1R agonist. Actually, it has high binding affinity to both CB1 and CB2 receptors with Ki values of 9 nM and 2.9 nM, respectively. As evidence has shown that hippocampal neurons also express functional CB2 receptors (referencing Stempel AV et al., 2016), it is necessary to evaluate whether pretreatment with CB1 or CB2 receptor antagonists will block the NAM action on JWH018-induced reduction in EPSCs.

A: One advantage of the autaptic model for the study of cannabinoid signaling is that we have had the opportunity (in 25+ papers) to extensively map out the molecular machinery that underlies DSE and other cannabinoid-related forms of plasticity and also to examine questions such as whether cannabinoid CB2 receptors might play a role regulating glutamate neurotransmission in these neurons.   We examined the question of CB2 function in the autaptic neuron model in Atwood et al., 2012 (PMC:22579668)).  There we demonstrated that while these neurons do not respond to CB2 agonists under baseline conditions, if they are transfected with CB2 they do develop CB2 responses that are qualitatively similar to those of CB1 receptor activation.   In the same year we also published Murataeva et al., 2012 (PMC:22921769) a study that indicates that a nominally CB2-selective agonist JWH015 is actually quite effective at CB1.   Regarding the work by Stempel et al, they report functional expression in CA3 neurons.  Our neuronal cultures are chiefly CA1 but do include a portion of the CA3.  It is possible that there is a subpopulation response that we have overlooked.   But given our extensive characterization of the model, we argue that CB2 does not contribute meaningfully to the regulation of neurotransmitter release as measured in DSE and that CB2 would not therefore contribute to the effects of JWH018 in the autaptic model that we report here. 

Reviewer 2 Report

Comments and Suggestions for Authors

1. Opiate does not mean opioid, as opiate refers to natural compounds extracted from the Papaverum, while the term "opioids" refers to all compounds both natural, synthetic, endo- and exogenous that interact with opioidergic receptors. Therefore, I suggest to correct opiate crisis into opioid crisis (abstract/introduction). 

2. Abstract: "spice" is not the only trade name for synthetic designer drugs being synthetic cannabinoids. 

3. Please provide the total number of the animals used in the study (methodology), including these used for hippocampal culture preparation.

4. In line with this, please indicate whether the animals were randomly divided or the Authors determined the differences in response between male and female mice.

5. As most of the cannabinoid drugs are difficult to dissolve in solvents such as saline, please indicate in which medium the test compounds were suspended/dissolved?

6.  On what basis did the Authors choose to use CB1 NAMs at a dose of 10 mg/kg i.p.; any dose-response for ABD or RTI compound? Also, I wonder whether the Authors posses any results for enzymatic stability of such compounds? it might be crucial, especially considering that the Authors choose to administer these compound 30 min before JWH018 

7. Please change the color of the curve for PSN compounds administered at different doses (Fig. 6D)

8. If possible, please provide some description for NAMs considering their structure which might help to understand differences in their activities

9. Please provide limitation of the study presented

Comments on the Quality of English Language

Minor changes are required

Author Response

  1. Opiate does not mean opioid, as opiate refers to natural compounds extracted from the Papaverum, while the term "opioids" refers to all compounds both natural, synthetic, endo- and exogenous that interact with opioidergic receptors. Therefore, I suggest to correct opiate crisis into opioid crisis (abstract/introduction). 
    1. We thank the reviewer for pointing this out. This has been corrected throughout the manuscript.
  2. Abstract: "spice" is not the only trade name for synthetic designer drugs being synthetic cannabinoids. 
    1. We were hoping to use Spice as a simple term for synthetic designer cannabinoids, but were aware that we might encounter resistance to this. We have substituted designer synthetic cannabinoid/SC in the text.    
  3. Please provide the total number of the animals used in the study (methodology), including these used for hippocampal culture preparation.
    1. Number of mice used has been added under Animals.
  4. In line with this, please indicate whether the animals were randomly divided or the Authors determined the differences in response between male and female mice.
    1. We apologize for this major oversight. The study made use of pups derived from either sex, but for behavioral experiments we made use of only male mice.  We did not use female mice because this is a proof-of-concept study to determine whether CB1 NAMs can, in principle, impact JWH018 responses in vitro and in vivo.  Moreover the majority of overdoses from synthetic cannabinoids are seen in males.   We have corrected this oversight in the text and added an explanation for why females were not tested.
  5. As most of the cannabinoid drugs are difficult to dissolve in solvents such as saline, please indicate in which medium the test compounds were suspended/dissolved?
    1. More details on solvents have been added to the methods section.
  6. On what basis did the Authors choose to use CB1 NAMs at a dose of 10 mg/kg i.p.; any dose-response for ABD or RTI compound? Also, I wonder whether the Authors posses any results for enzymatic stability of such compounds? it might be crucial, especially considering that the Authors choose to administer these compound 30 min before JWH018 
    1. A: We chose 10mg/kg partly based on the literature (e.g. Gamage et al., 2017 PMC5771238 for PSNCBAM1 and several RTICBM compounds), and partly based on our experience with lipophilic compounds targeting cannabinoid receptors. Presumably the reviewer is asking why we did not test higher concentrations.  We hesitate to make use of concentrations higher than 10mg/kg largely because of the risk of off-target effects.  We have some discussion of this in our study of cannabidiol (Straiker et al., 2018 PMID:29669714) which impacts a bewildering number of targets at higher concentrations.  For the second point, we do not have data on enzymatic stability for these compounds.  
  7. Please change the color of the curve for PSN compounds administered at different doses (Fig. 6D)
    1. The figure has been updated to make the difference in colors clearer.
  8. If possible, please provide some description for NAMs considering their structure which might help to understand differences in their activities
    1. We have added some discussion of how the NAM structures may relate to the observed effects (page 10)
  9. Please provide limitation of the study presented
    1. Our discussions of the changing nature of synthetic cannabinoids and the question of the most relevant behavior to test a candidate antidote each refer to limitations of the study.

Round 2

Reviewer 2 Report

Comments and Suggestions for Authors

the Authors have now improved the paper by giving answers to questions raised. 

I suggest this paper is ready to be published.